# The Validity and Reliability of the PHQ-9 and PHQ-2 on Screening for Major Depression in Spanish Speaking Immigrants in Chile: A Cross-Sectional Study

**DOI:** 10.3390/ijerph192113975

**Published:** 2022-10-27

**Authors:** Antonia Errazuriz, Rodrigo Beltrán, Rafael Torres, Alvaro Passi-Solar

**Affiliations:** 1Department of Psychiatry, School of Medicine, Pontificia Universidad Catolica de Chile, Santiago 8330077, Chile; 2Department of Public Health, School of Medicine, Pontificia Universidad Catolica de Chile, Santiago 8330077, Chile; 3Research Department of Epidemiology, Public Health University College London, London WC1E 7HB, UK

**Keywords:** depression, patient health questionnaire, Composite International Diagnostic Interview, population-based sample, immigrant mental health

## Abstract

Background: The study aimed to explore the psychometric properties of two versions of the Patient Health Questionnaires (PHQ-9 and PHQ-2) on screening for Major Depressive Disorder (MDD) among Spanish-speaking Latin American adult immigrants in Santiago, and to explore factors associated with a higher risk of occurrence of MDD among them. Methods: A representative sample of 897 Spanish-speaking immigrants completed the PHQ-9. The Composite International Diagnostic Interview (CIDI) was employed to evaluate MDD. Internal consistency and structural validity were evaluated using Cronbach’s α coefficient and confirmatory factor analysis (CFA). Convergent validity with the 7-item General Anxiety Disorder Scale (GAD-7) was assessed using Spearman’s correlations. Sensitivity, specificity, positive predictive values, and area under the receiver operating characteristic (ROC) curve were calculated for different cut-off points. Logistic regression analysis was used to identify factors associated with the risk of MDD. Results: Cronbach’s α coefficient of the PHQ-9 was 0.90; item-total correlation coefficients ranged from 0.61 to 0.76 and correlation with the GAD-7 was moderate (*r* = 0.625; *p* < 0.001). CFA on three alternative models suggests a plausible fit in the overall sample and among two of the subsamples: Peruvians and Venezuelans. Taking the results of CIDI as the gold standard for MDD, the area under the ROC curve was 0.91 (95% confidence interval (CI): 0.83~1.0). When the cut-off score was equal to 5, values of sensitivity, specificity, and Youden’s index were 0.85, 0.90, and 0.75, respectively. Multivariate logistic regression analyses showed that the influence of having three or more children (OR = 3.91, 95% CI: 1.20~12.81; *p* < 0.05), residency in Chile of up to three years (OR = 1.79, 95% CI: 1.07~3.00; *p* < 0.05), active debt (OR = 2.74, 95% CI: 1.60~4.70; *p* < 0.001), a one (OR = 2.01, 95% CI: 1.03~3.94; *p* < 0.05) and two or more events of adversity during childhood (OR = 5.25, 95% CI: 1.93~14.3; *p* < 0.01) on the occurrence of MDD was statistically significant. Reliability (α = 0.62), convergent (*r* = 0.534; *p* < 0.01) and criterion (AUC = 0.85, 95% CI: 0.67~1.00) validity coefficients of the PHQ-2 were weaker than for the PHQ-9. Conclusions: The PHQ-2 and the PHQ-9 are reliable and valid instruments for use as screeners for MDD among Spanish-speaking populations of Latin America.

## 1. Background

Major Depressive Disorder (MDD) is one of the most prevalent mental health conditions in the general population [1], globally affecting an estimated 4.7% of individuals in the community [2]. It is also the most common psychiatric condition in people who die from suicide [3] and one of the leading causes of burden [4]. MDD, however, is frequently untreated, particularly in low- and middle-income countries [5].

The nine-item Patient Health Questionnaire (PHQ-9) is one of the most widely used screening questionnaires for MDD [6] and has been translated into more than 80 languages [7]. Its accuracy has been shown to be greater than unaided clinician diagnoses with much higher sensitivity (0.80) and specificity (0.92) [8] than the sensitivity (0.50) and specificity (0.81) of primary care providers’ diagnoses [9]. 

The Spanish version of the PHQ-9 [10] has been validated in some Latin American populations. Its psychometric properties and convergent validity with a structured clinical interview as a reference gold standard have been documented in clinical settings in Argentina [11,12], Colombia [13], Peru [14], and Chile [15,16] and its factor structure has been studied among several non-clinical subpopulations: college students in Colombia [17], Ecuador [18] and Peru [19], and teachers [20] and rural communities in Mexico [21]. The performance of the first two items of the PHQ (i.e., PHQ-2) versus all nine items has also been assessed in primary care in Colombia [22] and among non-clinical populations in Chile [23] and Mexico [21].

Chile has been increasingly receiving economic immigrants in the last decade. In 2017, the estimated number of immigrants was 700,000, representing 4.4% of the total population [24]. A large majority of immigrants (76%) originate from Spanish-speaking Latin American countries, with Peruvians, Colombians, and Venezuelans forming the first, second, and third largest groups (50.4% of all immigrants [24]). While a lower prevalence of MDD among immigrants compared to the native-born population has been documented in high-income countries [25,26,27,28], evidence suggests that, in some cases, the migration experience may have a negative impact on mental health [29,30].

Because of the growth in the number of Latin American immigrants arriving in Chile, and the layers of vulnerability they experience [31], it is important to understand if the available screening measures for MDD are valid to use among them. 

To our knowledge, the Spanish version of the PHQ-9 has not been validated in non-clinical populations of Latin America with a reference gold standard and no studies have validated its screening characteristics among Spanish-speaking immigrants in Chile. To cover this gap, we designed the present study which analyses the psychometric properties of the PHQ-9 among a representative sample of native Spanish-speaking adult immigrants in Santiago, Chile. The primary aim focuses on the reliability of items (internal consistency), the dimensionality (structural validity), and convergence with a reference gold standard in identifying respondents with MDD (criterion validity). Secondary aims include (i) identifying factors associated with a higher risk of the occurrence of MDD in the Spanish-speaking immigrant population in Chile, and (ii) exploring the psychometric properties of the PHQ-2 among this population. The results of this study will provide evidence about the quality of this screening tool for MDD for this specific immigrant population as well as information about the prevalence of depressive symptoms and associated factors to MDD among them.

## 2. Material and Methods

### 2.1. Design

The Santiago Immigrant Wellbeing Study (STRING) is a population-based cross-sectional household mental health survey of 1115 first-generation adult immigrants residing in the Santiago Metropolitan Region (Región Metropolitana; RM) of Chile. The study was prospectively registered (ClinicalTrials.gov Identifier: NCT04114565; ISRCTN96875479) and approved by the Ethics Committee of the School of Medicine of the Pontificia Universidad Catolica de Chile (No. 170519004). Data was collected between August and October 2019.

### 2.2. Participants

The sampling framework of the Chilean National Institute of Statistics was used. Multi-stage random probability sampling, comparable to that of other household survey designs conducted in developing countries was employed [32]. Based on (i) the number of immigrants recorded in the RM in the 2017 Chilean Population Census [33], distributed in 120 conglomerates (i.e., clusters of 200 households with a mean number of 9.2 immigrants), and (ii) the density of the immigrant population in the RM, the minimum sample size to generate statistical inferences was set at 1104. The final sample was composed of 1115 participants, of which 1091 were interviewed in Spanish and 24 in Creole. To be eligible, participants had to: (i) be a community-residing adult (18+ years), (ii) be able to read and write, (iii) have been born outside of Chile (self-reported and providing a National ID number), and (iv) have been residing in Chile for at least six months.

This study aimed to explore the psychometric properties of the nine- and two-item Patient Health Questionnaires (i.e., PHQ-9 and PHQ-2) on screening for depression among the subsample of participants of the STRING study interviewed in Spanish and born in Spanish-speaking Latin American countries (n = 897).

### 2.3. Measures

The PHQ-9 is a nine-item self-reporting questionnaire that assesses the presence and severity of depressive symptoms (i. anhedonia, ii. depressed mood, iii. sleep problems, iv. low energy, v. appetite changes, vi. low self-esteem, vii. concentration difficulty, viii. psychomotor agitation or retardation and ix. suicidal ideation) in the last two weeks based on the DSM-IV criteria for MDD. Using a Likert scale, each item can be rated from 0 (not at all) to 3 (nearly every day) with total scores ranging from 0 to 27. Five categories of symptom severity have been proposed: 0–4 (minimum), 5–9 (mild), 10–14 (moderate), 15–19 (moderate to severe), 20–27 (serious) [6] and cut-off scores between 8 and 11 have been recommended for a probable case of MDD [34]. In this study, the Spanish version of the questionnaire was employed [10] and administered face-to-face.

The PHQ-2 is an ultra-brief version of the PHQ-9 which includes the first two items (i. anhedonia, ii. depressed mood). Total scores range from 0 to 6 and the recommended cut-off point for administering the full PHQ-9 or conducting a clinical interview to assess for MDD is a score of 3 or greater [35]. 

Symptoms of anxiety during the 2 weeks preceding the interview were assessed using the Spanish version of the 7-item General Anxiety Disorder (GAD–7) scale [36]. DSM-IV MDD in the last 12 months was assessed using two modules of the WHO Composite International Diagnostic Interview (WHO-CIDI) (CAPI 3.0): depression, and mania [37]. The WHO-CIDI is a gold standard for MDD diagnosis which has been used extensively in major epidemiological studies in Argentina [38], Chile [39,40], Colombia [5,41,42,43,44,45,46,47], Guatemala [48], Mexico [5,49,50,51] and Peru [5,52]. 

Childhood trauma was measured using the Spanish-translated version of the Adverse Childhood Experiences International Questionnaire (ACE-IQ) [53] and coded into 0, 1, or 2 or more adverse events.

Participants’ gender, age, country of birth, number of children aged under 18 years, duration of residency in Chile (up to 3 years; more than 3 years), highest educational level (primary: 8 years or less; secondary: 9 to 12 years; or higher: >12 years), employment status (unemployed or economically inactive versus employed), active debt status (with or without) was recorded.

### 2.4. Statistical Analysis

To characterize the representative immigrant population of Santiago, sampling weights assigned to the subjects were applied. For descriptive statistics, means, standard deviations, and frequencies were calculated for demographic and economic factors and for the number of childhood adversity events. 

The normality distribution was tested using the Kolmogorov-Smirnov test. When variables were not found to have a normal distribution, Spearman’s Rho was used. For reliability, the internal consistency of the PHQ-9 was assessed using Cronbach’s alpha (α) coefficients and item-total correlations. For the PHQ-2, Spearman–Brown coefficients were calculated as an additional measure of reliability. Alpha coefficients above 0.70–0.80 were considered indicative of good internal consistency [54]. Correlations above 0.90, between 0.70 and 0.90, and between 0.50 and 0.70 were considered very high, high, and moderate, respectively [55]. 

To investigate the factor structure of the PHQ-9 items, Confirmatory factor analysis (CFA) with the maximum-likelihood procedure was conducted. Using CFA, three alternative models of the structure of the PHQ-9 were tested to understand the dimensionality of the scale: (i) the original one-factor model hypothesized by Kroenke [6], (ii) a two-factor model suggested by Krause [56] where three items (i.e., ‘sleep problems’, ‘low energy’ and ‘appetite changes’) are loaded on a somatic factor, and (iii) another two-factor model derived by Richardson and Richard [57] where the somatic factor also includes ‘concentration difficulties’ and ‘psychomotor agitation/retardation’ (Appendix A). In addition to the chi-square (χ^2^) and its degrees of freedom (df), models were compared using the following three model fit indices and their criteria: (i) the robust comparative fit index (CFI) with a value greater than 0.95 indicating a good model fit and values larger than 0.90 indicating a plausible model fit [58], (ii) the robust root mean square error of approximation (RMSEA) and its 90% confidence interval with a value lower than 0.05 indicating a good fit and values lower than 0.08 indicating a plausible fit [59], and (iii) the standardized root mean square residual (SRMR) with values less than 0.05 indicating a good fit well and values lower than 0.08 indicating a plausible fit [60]. A model was considered to fit well if two of the three criteria were met.

The convergent validity of the PHQ-2 and PHQ-9 was assessed using Spearman’s correlations with the GAD-7. To assess the accuracy of both forms of the PHQ as screening tools compared to the CIDI, the receiver operating characteristics (ROC) and AUC were analyzed. The optimum cut-off point for the PHQ-9 was determined considering validity indices: sensitivity (Se), specificity (Sp), positive and negative predictive value (PPV; NPV), Youden’s index, positive and negative likelihood ratios (LR), and ROC curve/AUC analysis. Two additional indices were used: clinically relevant rule in accuracy as estimated by the clinical utility index positive (CUI+) and rule out accuracy as estimated by the clinical utility index negative (CUI) [61]. 

To evaluate the association between demographic, economic, and adversity characteristics and a total PHQ-9 total score over the defined optimum cut-off point, age, and gender-adjusted logistic regression analyses were conducted.

The statistical analysis, excluding CFA, was performed using SPSS for Windows, version 20.0 (IBM Corp, Armonk, NY, USA). CFA was conducted using R 3.5.1 software (The R Foundation for Statistical Computing, Vienna, Austria). A *p*-value < 0.05 was considered significant.

## 3. Results

### 3.1. Sample Characteristics

Table 1 illustrates the sociodemographic, economic, and childhood adversity characteristics of the final sample (n = 897). The mean age of participants was 36.6 years (SD = 11.5); 478 (53.6%) had no children under the age of 18, 217 (22.7%) had one, 144 (16.2%) had two and 55 (7.5%) had three or more children. There were 509 women (53.8%) and 388 men (46.2%); 94 respondents had been born in Colombia (13.2%), 333 in Peru (26.6%), 375 in Venezuela (33.3%), and 95 (26.9%) in other Spanish-speaking Latin American countries (Bolivia, Cuba, Dominican Republic, Ecuador, and Mexico); 444 (50.3%) participants had migrated in the last three years and 453 (49.7%) more than three years ago. 

Few participants (7.6%) were unemployed or economically inactive; 72 (7.2%) respondents had ≤8 years of education, 351 (41.6%) had between 9 and 12, and 471 (51.2%) had more than 12 years, and 201 (22.4%) held at least one debt at the moment of the interview. No events of childhood adversity were reported by most participants (68.6%), one event was reported by 15.7%, and two or more events by 15.7% of participants.

Only 34 (4.5%) participants scored above the PHQ-9 most common cut-off score (≥10), 42 (4.6%) respondents above the recommended cut-off score for the PHQ-2 (≥3), and only 6 (0.7%) participants met the criteria for DSM-IV MDD in the last 12 months (Table 2). Results for specific immigrant groups showed the proportion of PHQ-9 total scores ≥ 10 ranged from 1.7% among Venezuelans to 7.1% among Peruvians, and the proportion of PHQ-2 scores ≥ 3 ranged from 1.4% among other Latin Americans to 8.1% among Colombians (Appendix A). 

### 3.2. Reliability

Values of the reliability coefficient Cronbach’s α for the PHQ-9 and PHQ-2 were 0.904 and 0.746, indicating acceptable reliability. For both PHQ forms, corrected item-total correlations ranged from moderate to high (0.604–0.760), and the internal consistency of the PHQ-9 would not have improved with the deletion of later scale items. The Spearman–Brown between-item correlation coefficient for the PHQ-2 was significant but moderate (*r* = 0.62; *p* < 0.001) (Appendix A). The reliability of the PHQ-9 was acceptable across immigrant groups (all α > 0.7) and corrected item-total correlations ranged from low to high (0.451–0.845) (Appendix A). The reliability of the PHQ-2 was only acceptable for the Peruvian and other Latin American subsamples (all α > 0.7 and *r* > 0.70; *p* < 0.001) (Appendix A). 

### 3.3. Factor Structure

Confirmatory factor analysis (CFA) on three alternative models was performed: a one-factor model (model 1) and two two-factor models (models 2a and 2b). The goodness of fit indices suggest a plausible fit for the three models for the overall sample and the Peruvian subsample and for model 2a for the Venezuelan subsample, with CFI values larger than 0.90 and SRMS values lower than 0.08. The goodness of fit indices for the Colombian and the other Latin American subsamples demonstrate inadequate fit of all three models (Appendix A). Overall results from the two-factor models showed that the somatic and affective factors were highly correlated (all *p* < 0.001). 

### 3.4. Convergent Validity

In the overall sample, the PHQ-9 was moderately and positively correlated with the GAD-7 (*r* = 0.625; *p* < 0.001), which measures anxiety symptoms. Correlation between PHQ-2 and GAD-7 scores was weaker, but significant (*r* = 0.534; *p* < 0.01). Across immigrant groups, correlations between the GAD-7 and the PHQ-9 and PHQ-2 were moderate (all *r* ≥ 0.60; *p* < 0.001) and low-to-moderate (all *r* ≥ 0.44; *p* < 0.001), respectively. Convergent validity was, therefore, satisfied for both forms of the PHQ with our sample (Appendix A).

### 3.5. Criterion Validity

The performance of the two versions of the PHQ was examined against the diagnosis of MDD according to the WHO-CIDI as a gold standard. The mean PHQ-9 and PHQ-2 scores for participants meeting the criteria for MDD were 9.14 (SD = 5.36) and 2.19 (SD = 1.39), respectively, whereas the mean scores for participants not meeting the criteria for MDD were 1.71 (SD = 3.68) and 0.38 (SD = 0.98), respectively.

Table 3 summarizes the operating characteristics of the PHQ-9 and PHQ-2 at different cut-off points for diagnosing MDD against the WHO-CIDI. For the PHQ-9, the cut-off of 5 yielded the best diagnostic performance in terms of maximizing sensitivity and specificity values based on the Youden J index (sensitivity = 85%; specificity 90%). Despite the excellent rule-out accuracy (CU− = 0.90) and NPV (NPV = 1.00) for this cut-off score, negative and positive likelihood ratios were moderate (LR+ = 8.73; LR− = 0.17) and PPV and rule-in accuracy were very poor (PPV = 0.06; CUI+ = 0.05). The ROC curve analysis estimated the area under the curve (AUC) for the PHQ-9 was 0.91 (95% CI: 0.83–1.00) (Figure 1), which accounts for outstanding classification accuracy.

For the PHQ-2, optimal diagnostic performance was achieved at a cut-off score of 1, which best balances sensitivity and specificity (sensitivity = 73%; specificity 89%). Despite the high NPV (NPV = 1.00) and excellent rule-out accuracy (CU− = 0.89) for this cut-off score, likelihood ratios were moderate (LR+ = 6.58; LR− = 0.30) and PPV and rule-in accuracy were very poor (PPV = 0.04; CUI+ = 0.89). The AUC of PHQ-2 was 0.85 (95% CI: 0.67–1.00) which accounts for excellent accuracy [62].

### 3.6. Variables Associated with PHQ-9 Scores over the Optimal Cut-Off Score

As presented in Table 4, a total PHQ-9 score over the optimal cut-off score (i.e., ≥5) was strongly associated with the experience of adversity during childhood. Compared to participants reporting no childhood adversity, those reporting one or two or more events of adversity were twice (OR 2.01, 95% CI 1.03–3.94) and five times (OR 5.25, 95% CI 1.93–14.34) more likely to score ≥ 5 in the PHQ-9, respectively. A PHQ-9 score ≥ 5 was also associated with length of residency, active debt, and the number of children. The odds of reporting a score above the cut-off were higher among those who had immigrated in the last three years (OR 1.79, 95% CI 1.07–3.00) versus those who had been residing in Chile for over three years, among those reporting active debt (OR 2.74, 95% CI 1.07–3.00) and among those with three or more children (OR 3.91, 95% CI 1.20–12.81).

## 4. Discussion

The present study is the first to analyze the psychometric properties of the PHQ-9 and PHQ-2 in a representative sample of Spanish-speaking adult immigrants in Chile. Its results expand the evidence of the validity, reliability, and accuracy of both versions of the PHQ to a specific non-clinical population and contribute to the literature about the prevalence of depressive and anxiety symptoms and associated factors among economic immigrants. 

Consistent with findings from available primary care studies in Chile [15,16,64,65], good internal consistency of the PHQ-9 in the overall sample and across immigrant groups was demonstrated. As expected [66], the correlation between the PHQ-9 with the GAD-7 was significant but moderate indicating convergent validity and supporting the growing body of literature about the coexistence and comorbidity of depression and anxiety [67,68]. 

Contrary to Latin American validation studies of the PHQ-9 among clinical [16,66,69] and non-clinical populations [17,19,20,21], which have demonstrated a good fit for both one- and two-factor solutions using confirmatory factor analysis, the goodness of fit indices for the three models tested in this study only indicate a plausible fit in the overall sample and in two of the four subsamples: Peruvians and Venezuelans. Some studies in the US have reported a poor fit of the PHQ-9 among Hispanic participants [70] but the pattern of a plausible fit across models in the overall sample and an inadequate fit in the Colombian and other Latin American subsamples observed in this study indicate the need to expand research about the structure of the Spanish version of the PHQ-9 among non-clinical populations in Latin America.

The PHQ-9 demonstrated high diagnostic accuracy and adequate sensibility and specificity values, comparable to those reported in studies conducted in clinical populations in Chile using the CIDI as a gold standard [15,16]. As expected of a screener for a low prevalence condition, it performed better at correctly identifying individuals without MDD than at confirming the presence of MDD, with negative (NPV) and positive (PPV) predictive values for scores between 5 and 10 in the range of 0.99–1.00 and 0.06–0.04, respectively. The low PPV suggests that rather than indicate a probable MDD diagnosis, a high PHQ-9 score should be regarded as an indication of the need for a more thorough evaluation. 

The optimal PHQ-9 cut-off score for detecting MDD in the present study was ≥5. This score is lower than the original cut-off score (≥10) [6]. It is also lower than reported in a recent individual participant data meta-analysis (IPDMA) based on data from 100 studies which concluded that a cut-off score of ≥8 maximized combined sensitivity and specificity for fully structured reference standards [71]. However, the comparison of these values is limited considering the questionnaire was originally developed using data from primary care [6], and the number of studies from non-clinical samples included in the recent meta-analysis is low (15%) [71]. 

Across parameters and subsamples, the PHQ-9 demonstrated better performance than the PHQ-2: higher internal consistency, stronger convergent validity with the GAD-7, better accuracy with regard to a gold standard, and higher sensitivity and specificity for the proposed cut-off scores. However, taking into account the brevity of the PHQ-2 and in consistency with findings from studies among clinical [22] and non-clinical [21] populations in Latin America, results from this study suggest its performance is acceptable.

Findings from this study suggest that the prevalence of depressive symptoms and MDD among and across Spanish-speaking immigrant groups is lower than among the native population of Chile and other Spanish-speaking Latin American countries, independent of the assessment method used. Results from general population studies using structured clinical interviews in Chile [39,40,72], Colombia [5,42,43], and Peru [73] have consistently reported a higher prevalence of MDD than observed in the present study. Results from a general population study conducted in Chile using the PHQ-9 [23] also report a higher prevalence of MDD than reported in the current study. This lower risk for MDD represents a mental health advantage similar to that observed among first-generation immigrants from Latin America in the United States [74] and may partly be explained by their high labor force participation and low level of exposure to adversity during childhood.

Four factors predicted an increased risk of MDD among immigrants: the number of children (≤18 yrs), short length of residency in Chile (≤3 yrs), active debt, and exposure to childhood adversity. Contrary to evidence suggesting that an increase in the length of residency in the host country is associated with a higher risk of MDD among immigrants [75], the risk of MDD in this immigrant population was not predicted by a longer duration of residency. This lack of finding may be related to (i) the small proportion of participants who have been exposed to the prolonged effects of migration (only 12% have been residing in Chile for over 10 years), or (ii) the characteristics of the acculturation process shaped by high levels of cultural contiguity between Chile and other Latin American Spanish-speaking countries at the linguistic, religious, and ethnic level. Findings from this study support the well-established relationships between unsecured debt and mental health [76] and between adversity during childhood [77], having three or more children [78], and depression in adulthood.

Several limitations of this study should be acknowledged. First, the generalizability of the findings regarding the psychometric properties of both versions of the PHQ to non-native Spanish-speaking immigrants may be limited. Second, the generalizability regarding the prevalence of depressive symptoms and MDD and associated factors to non-economic migrants, migrants from high-income countries, from outside the Latin American region, or refugees may be limited because of differential exposure to stress throughout the migration process. Replication with more heterogeneous samples of immigrants such as individuals with a longer duration of residency in Chile, with clinical conditions, or service users is thus needed.

## 5. Conclusions

Findings from the present study indicate that both the PHQ-2 and PHQ-9 are psychometrically sound measures of depression for use among Spanish-speaking populations in Latin America. These results add to the evidence that they are brief, easy to use, and valid depression measures for use in clinical and non-clinical settings.

## Figures and Tables

**Figure 1 ijerph-19-13975-f001:**
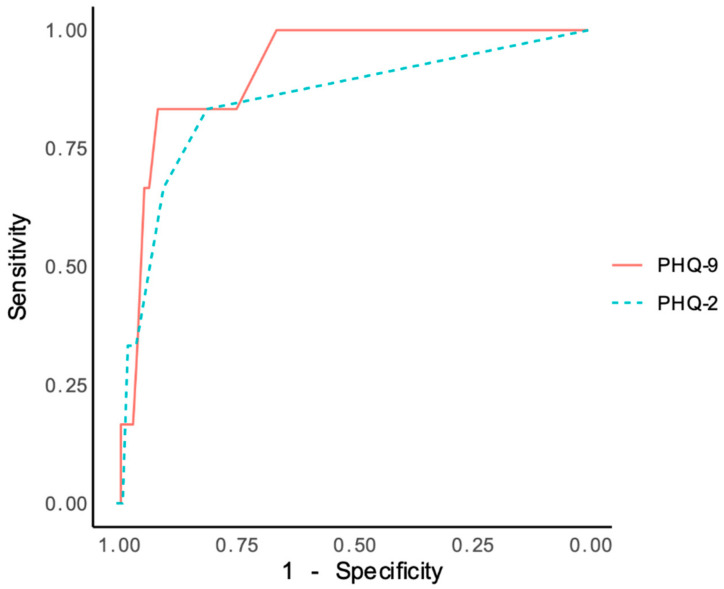
Receptor operating characteristics (ROC) curves of the PHQ-9 and PHQ-2 compared with the Composite International Diagnostic Interview as a reference standard for MDD.

**Table 1 ijerph-19-13975-t001:** Sociodemographic, economic, and childhood trauma characteristics of participants (n = 897).

			n	(%)
Sociodemographic				
	Gender			
		Female	509	(53.8)
		Male	388	(46.2)
	Age			
		18 to 27 years	199	(25.6)
		28 to 37 years	349	(39.4)
		38 to 47 years	184	(17.8)
		48 or + years	165	(17.2)
	Number of children (<18)			
	0		478	(53.6)
	1		217	(22.7)
	2		144	(16.2)
	3 or more		55	(7.5)
	Country of birth			
		Colombia	94	(13.2)
		Peru	333	(26.6)
		Venezuela	375	(33.3)
		Other LA	95	(26.9)
	Duration of residency in Chile			
		Up to 3 years	444	(50.3)
		More than 3 years	453	(49.7)
Economic				
	Education			
		Primary (≤8 yrs)	72	(7.2)
		Secondary (9–12 yrs)	351	(41.6)
		Higher (>12 yrs)	471	(51.2)
	Employed, yes (ref: unemployed/inactive)		823	(92.4)
	Debt status, with (ref: without)		201	(22.4)
Childhood trauma				
	Events of adversity			
		0	639	(68.6)
		1	137	(15.7)
		2 or more	121	(15.7)

Note: Number of cases unweighted and % weighted; Latin America (LA).

**Table 2 ijerph-19-13975-t002:** Distribution of PHQ scores and 12-month-CIDI Major Depressive Disorder (n = 897).

Questionnaire			CIDI	
Depressive Symptoms Severity	n	(%)	Negative	Positive
PHQ-9				
None (0–4)	790	(86.8)	789	1
Mild (5–9)	73	(8.6)	69	4
Moderate (10–14)	19	(2.3)	19	0
Severe (≥15)	15	(2.2)	14	1
PHQ-2				
Minimum (0–2)	855	(95.4)	851	4
Mild (≥3)	42	(4.6)	40	2
Total	897	(100.0)	891	6

Note: Number of cases unweighted and % weighted. Composite International Diagnostic Interview (CIDI); 9-item Patient Health Questionnaire (PHQ-9); 2-item Patient Health Questionnaire (PHQ-2).

**Table 3 ijerph-19-13975-t003:** Performance of PHQ-9 and PHQ-2 cut-off scores in detecting major depressive disorder.

Scale	Cut-Point	Sensitivity	Specificity	Youden’s Index (J)	PPV	NPV	CUI+	CUI−	LR+	LR−
PHQ-9										
	PHQ ≥ 1	0.85	0.72	0.57	0.02	1.00	0.02	0.72	3.07	0.21
	PHQ ≥ 2	0.85	0.80	0.64	0.03	1.00	0.02	0.79	4.14	0.19
	PHQ ≥ 3	0.85	0.84	0.69	0.04	1.00	0.03	0.84	5.43	0.18
	PHQ ≥ 4	0.85	0.87	0.72	0.04	1.00	0.04	0.87	6.65	0.18
	PHQ ≥ 5	0.85	0.90	0.75	0.06	1.00	0.05	0.90	8.73	0.17
	PHQ ≥ 6	0.73	0.92	0.65	0.06	1.00	0.04	0.92	9.51	0.29
	PHQ ≥ 7	0.73	0.93	0.66	0.07	1.00	0.05	0.93	10.81	0.29
	PHQ ≥ 8	0.49	0.95	0.43	0.06	1.00	0.03	0.94	8.92	0.54
	PHQ ≥ 9	0.22	0.96	0.17	0.03	0.99	0.01	0.95	4.89	0.82
	PHQ ≥ 10 ^a^	0.22	0.96	0.18	0.04	0.99	0.01	0.96	5.96	0.81
	PHQ ≥ 11	0.22	0.97	0.19	0.05	0.99	0.01	0.97	7.59	0.81
	PHQ ≥ 12	0.22	0.97	0.19	0.05	0.99	0.01	0.97	7.59	0.81
	PHQ ≥ 13	0.22	0.97	0.19	0.05	0.99	0.01	0.97	7.80	0.81
	PHQ ≥ 14	0.22	0.98	0.20	0.06	0.99	0.01	0.97	10.27	0.80
	PHQ ≥ 15	0.22	0.99	0.21	0.14	0.99	0.03	0.99	24.1	0.79
PHQ-2										
	PHQ ≥ 1	0.73	0.89	0.62	0.04	1.00	0.03	0.89	6.58	0.30
	PHQ ≥ 2	0.31	0.96	0.26	0.04	1.00	0.01	0.95	7.00	0.72
	PHQ ≥ 3 ^b^	0.31	0.98	0.29	0.08	1.00	0.03	0.97	13.76	0.71

Note: Clinical utility index positive (CUI+); Clinical utility index negative (CUI−); Likelihood Ratio positive (LR+); Likelihood Ratio negative (LR−); Negative Predictive Value (NPV); 9-item Patient Health Questionnaire (PHQ-9); 2-item Patient Health Questionnaire (PHQ-2); Positive Predictive Value (PPV). ^a^ A cut-off of 10 is recommended for the PHQ-9 [6]. ^b^ A cut-off of 3 is recommended for the PHQ-2 [63].

**Table 4 ijerph-19-13975-t004:** Variables independently associated with PHQ-9 ≥ 5 with respective adjusted odds ratios, 95% confidence intervals, and *p*-values.

			PHQ-9 ≥ 5		
Factor			OR	95% CI	*p*-Value
Sociodemographic					
	Gender ^a^				
		Male	1		
		Female	1.33	0.54 to 3.30	0.541
	Age group ^b^				
		18–27	1		
		28–37	0.63	0.37 to 1.06	0.083
		38–47	0.63	0.30 to 1.35	0.240
		≥48	0.91	0.48 to 1.71	0.762
	Number of children (<18)				
	0		1		
	1		1.62	0.89 to 2.95	0.117
	2		1.54	0.72 to 3.30	0.265
	3 or more		3.91	1.20 to 12.81	0.027
	Country of birth ^c^				
		Venezuela	1		
		Colombia	0.85	0.32 to 2.24	0.743
		Perú	1.47	0.77 to 2.82	0.247
		Other LA	1.00	0.25 to 4.04	0.998
	Duration of residency ^c^				
		>3 yrs	1		
		≤3 yrs	1.79	1.07 to 3.00	0.029
Economic					
	Education ^c^				
		Higher (>12 yrs)	1		
		Secondary (9–12 yrs)	1.94	0.74 to 5.09	0.181
		Primary (≤8 yrs)	2.41	1.01 to 5.75	0.053
	Currently holding debt ^c^				
		No	1		
		Yes	2.74	1.60 to 4.70	<0.001
Childhood Adversity					
	Events (no.) ^c^				
		0	1		
		1	2.01	1.03 to 3.94	0.045
		≥2	5.25	1.93 to 14.34	0.002

^a^ Adjusted by age in years. ^b^ Adjusted by gender. ^c^ Adjusted by age in years and gender.

## Data Availability

The datasets generated during and/or analysed during the current study are available from the corresponding author on reasonable request.

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
