# Peer review of "The Validity and Reliability of the PHQ-9 and PHQ-2 on Screening for Major Depression in Spanish Speaking Immigrants in Chile: A Cross-Sectional Study"

_ijerph, 2022, doi:10.3390/ijerph192113975_

Round 1
Reviewer 1 Report
Dear Authors,
The introduction is good and clarifies the need for the study and its application to the reader.
In the methods section, it is recommended to include the dates of the evaluations. Regarding the inclusion criteria, how did the authors verify the data of the immigrants, were the immigrants registered in any official database in Chile?
It is suggested that the authors explain how they arrive at a minimum sample size of participants to generate statistical inferences.
The authors state that they used the Spanish version of the questionnaire. Was the Spanish version validated in Chile and other Latin American countries?
Why did the authors decide to apply the questionnaire by telephone? How did they obtain the telephone numbers of the immigrants? In this sense, explain how they obtained the participants' informed consent.
It is suggested to change the word sex for gender.
Please justify the age categories used in the research.
For Cronbach's alpha, please indicate the cut-off points that you will consider as excellent (indicate references). The same suggestion applies to the correlations. Then you describe the results but without first indicating the cut-off points.
The results are ok. In addition, logistic regression incorporates essential information about the participants. In this sense, you have sociodemographic information such as marital status and number of children to be incorporated into the model. Also, did you evaluate the model by country of birth? My comment points out that immigrants may have different problems depending on their origin (in the supplements, you incorporate separate tables by country).
Finally, the results are fascinating and relevant to your country's migration policies.
Author Response
We thank the reviewer for their favorable evaluation of our study’s contribution.
Comment 1: In the methods section, it is recommended to include the dates of the evaluations.
Author response: We included data collection dates (see lines 105-106).
Comment 2: Regarding the inclusion criteria, how did the authors verify the data of the immigrants, were the immigrants registered in any official database in Chile?
Author response: Participants were asked to report their place of birth (city and country) and provide their national identification number (i.e. RUT). Data were not matched to an official database. However, the RUT is a unique sequential number given at birth to Chilean nationals. RUT values range from 1 to 25,000,000 and since higher numbers indicate a more recent date of birth, adults over 18 hold numbers under 18,000,000. Just like newborns, foreigners are granted a RUT number when applying for a visa. Considering that only 61 participants (6.8%) reported a length of residency of over 17 years, the large majority of participants reporting a place of birth outside of Chile held RUT numbers over 18,000,000 making their immigrant status inferable based on this number. We have added a comment about the self-reported nature of immigrant status in section 2.2 (see lines 118-119).
Comment 3: It is suggested that the authors explain how they arrive at a minimum sample size of participants to generate statistical inferences.
Author response: Criteria for calculating the minimum sample size was detailed in section 2.2 (see lines 111-116).
Comment 4: The authors state that they used the Spanish version of the questionnaire. Was the Spanish version validated in Chile and other Latin American countries?
Author response: the Spanish version of the PHQ has been validated in clinical populations in Argentina, Colombia, Peru and Chile (see lines 65-66).
Comment 5: Why did the authors decide to apply the questionnaire by telephone? How did they obtain the telephone numbers of the immigrants?
Author response: the questionnaire was applied in a face-to-face interview. By mistake, in the submitted version of the manuscript, the authors had reported it was applied by telephone (see line 134).
Comment 6: In this sense, explain how they obtained the participants' informed consent.
Author response: informed consent was obtained at participants' homes, prior to initiating the survey interview (see line 441).
Comment 7: It is suggested to change the word sex for gender.
Author response: the word sex has been changed for gender throughout the manuscript.
Comment 8: Please justify the age categories used in the research.
Author response: age group were based on 10-year ranges starting at the minimum age of participants (i.e. 18 years). Because of the low number of immigrants ages ‘58 or more’ (n=55), we decided to group the ‘48 to 57’ age group (n=110) with the 58 or more to produce the ‘48 or more’ category.
Comment 9: For Cronbach's alpha, please indicate the cut-off points that you will consider as excellent (indicate references).
Author response: values for interpreting the size of alpha coefficients and correlations were specified and referenced in section 2.4 (see lines 166-169).
Comment 10: The same suggestion applies to the correlations. Then you describe the results but without first indicating the cut-off points.
Author response: based on specified values for interpreting the size of alpha coefficients and correlations, results were updated for consistency in sections 3.2 (see lines 243-246) and 3.4 (see lines 260-261).
Comment 11: In this sense, you have sociodemographic information such as marital status and the number of children to be incorporated into the model.
Author response: information on marital status was available but since marital status was not independently associated with a score of 5 or more in the PHQ-9 ≥ 5, we did not include it in the sample description (i.e. Table 1) nor in the model (i.e. Table 4). Information on number of children (<18) has now been included in Tables 1 (see line 218) and 4 and discussed in the text (see lines 325-326; 397-398; 412-413).
Comment 12: Also, did you evaluate the model by country of birth? My comment points out that immigrants may have different problems depending on their origin (in the supplements, you incorporate separate tables by country).
Author response: Considering the small sample size of the immigrant groups (n=94–375) and that identifying factors associated with a higher risk of occurrence of MDD was a secondary aim, we did not test for associations between sociodemographic, economic and childhood adversity factors and PHQ-9 scores over the optimal cut-off score in specific immigrant groups.
Reviewer 2 Report
Dear authors,
I would like to congratulate you for your very interesting and well written work.
I have some small comments: a) One of your aim was to explore the psychometric properties of the PHQ-2 among the population but you don't make a relative reference for your findings in abstract section (in conclusion) and also in the end of the discussion. In my opinion, you must write something about your findings. b) The section "Discussion and Conclusion" must be divided in two separate section. Also, you could add more information in the conclusion.
Your research work is valuable.
Author Response
We thank the reviewer for their favorable evaluation of our study’s contribution.
Comment 1: One of your aims was to explore the psychometric properties of the PHQ-2 among the population but you don't make a relative reference for your findings in abstract section (in conclusion) and also in the end of the discussion. In my opinion, you must write something about your findings.
Author response: In the abstract, the performance of the PHQ-2 is described (see lines 32-34) and is now included in the conclusion section (see lines 34-35). In the discussion section, the context under which its performance is considered acceptable was added (see line 381).
Comment 2: The section "Discussion and Conclusion" must be divided in two separate sections. Also, you could add more information in the conclusion.
Author response: The "Discussion and Conclusion" section was divided into two and information about the performance of the PHQ-2 was included (see lines 425-428).
Reviewer 3 Report
This paper is well-written and covers a familiar yet interesting topic.
The methodology is detailed and supports the findings of the study.
Author Response
We thank the reviewer for their favorable evaluation of our study’s contribution.